# Dissociable effects of emotional stimuli on electrophysiological indices of time and decision-making

**Keri Anne Gladhill**[1]*, **Giovanna Mioni**[2], **Martin Wiener**[1]

**1** Psychology Department, George Mason University, Fairfax, Virginia, United States of America,
**2** Department of General Psychology, University of Padova, Padova, Italy

\* kgladhil@gmu.edu

**Data Availability Statement:** Data are available through OSF. Here is the DOI for all associated data. DOI: 10.17605/OSF.IO/NXFEP.

**Funding:** The author(s) received no specific funding for this work.

## Abstract

Previous research has demonstrated that emotional faces affect time perception, however, the underlying mechanisms are not fully understood. Earlier attempts focus on effects at the different stages of the pacemaker-accumulator model (clock, memory, and/or decision-making) including, an increase in pacemaker rate or accumulation rate via arousal or attention, respectively, or by biasing decision-making. A visual temporal bisection task with sub-second intervals was conducted in two groups to further investigate these effects; one group was strictly behavioral whereas the second included a 64-channel electroencephalogram (EEG). To separate the influence of face and timing responses, participants timed a visual stimulus, temporally flanked (before and after) by two faces, either negative or neutral, creating three trial-types: Neg→Neut, Neut→Neg, or Neut→Neut. We found a leftward shift in bisection point (BP) in Neg→Neut relative to Neut→Neut suggests an overestimation of the temporal stimulus when preceded by a negative face. Neurally, we found the face-responsive N170 was larger for negative faces and the N1 and contingent negative variation (CNV) were larger when the temporal stimulus was preceded by a negative face. Additionally, there was an interaction effect between condition and response for the late positive component of timing (LPCt) and a significant difference between response (short/long) in the neutral condition. We concluded that a preceding negative face affects the clock stage leading to more pulses being accumulated, either through attention or arousal, as indexed by a larger N1, CNV, and N170; whereas viewing a negative face after impacted decision-making mechanisms, as evidenced by the LPCt.

## Introduction

The ability to perceive time accurately is invaluable to humans. It is important in everyday activities to understand when in time and for how long an event occurred, or how soon in the future an event might occur. These different aspects of time allow us to move through the world and perform necessary actions on time. It is necessary then that we understand the conditions under which time perception can be altered and the inaccuracies that result. Emotion

**Competing interests:** The authors have declared that no competing interests exist.

and the passing of time are closely linked in the human language; for example, we often comment on time passing more quickly when having fun and time passing more slowly during distressing events. Given this close link it is not surprising that emotion has been found to alter one's perception of time [1]. Multiple theories have been presented to explain how emotion alters time perception; the focus of the current study is to tease apart differing explanations between them and compare contributing factors [2].

The pacemaker-accumulator model of time perception, or scalar expectancy theory (SET), first postulated by Treisman [3], and later expanded by Gibbon, Church, and Meck [4], includes three main stages: the clock, memory, and decision-making (Fig 1). The clock stage involves a pacemaker which constantly produces pulses. At the onset of a to-be-timed event, pulses are collected by an accumulator at a rate that is modulated by a switch- or gate-like mechanism [5–7]. The memory stage involves comparing the duration of an interval based on the number of pulses collected in working memory to previously stored values in long-term memory. Last is the decision-making stage which involves making a judgment regarding the previously made comparison. This decision can be made by comparing a temporal stimulus with another of a different duration through the reproduction of a time interval or by determining when to perform an action [1, 8].

Emotion has been proposed to impact time by acting on distinct stages in the pacemaker accumulator model. First, an increase in arousal by an emotionally arousing stimuli will increase the rate of the pacemaker which in turn leads to a larger number of pulses accumulating leading to a longer perception of time. For example, emotional states that involve higher levels of arousal (e.g., anger or fear) increases the pacemaker rate more compared to emotions with lower levels of arousal (e.g., sad). However, emotions with lower levels of arousal would still increase the rate of the pacemaker beyond that of neutral emotions [9]; therefore, more arousing stimuli will lead to a larger dilation of time than less arousing stimuli.

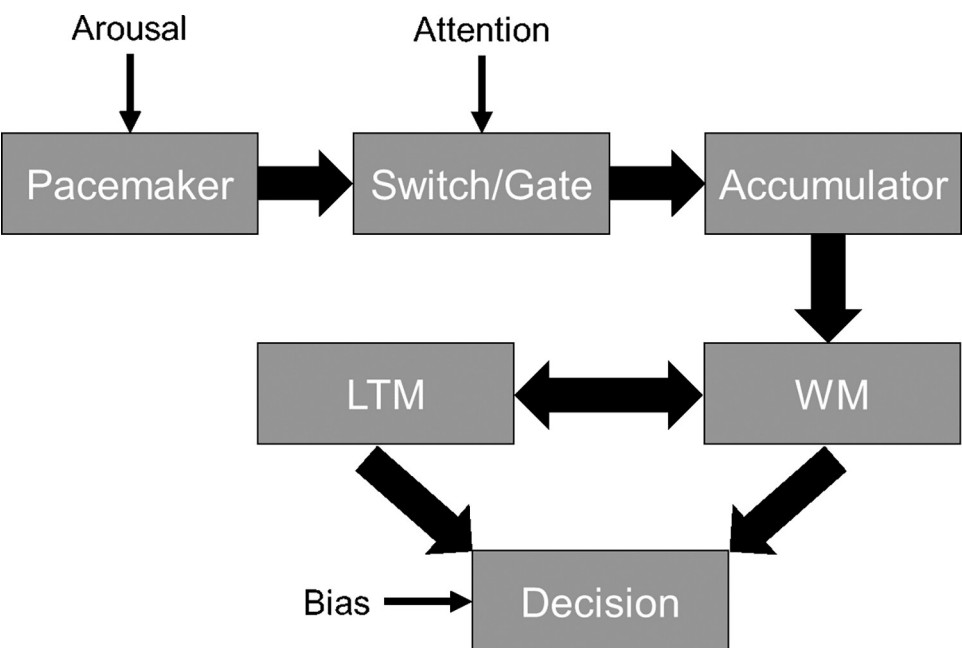

**Fig 1. Pacemaker-accumulator model of time perception/scaler expectancy theory (SET).** Theory includes three main stages: the clock, memory, and decision-making. Emotion may affect arousal via the pacemaker, attention via the switch/gate, and/or bias the decision (for review see: [5].

Second, changes in attention due to emotional stimuli can alter the distribution of cognitive resources to or away from a timing task thus affecting the gate or switch function of the model [1]. Lui and colleagues [10] suggest an emotional temporal stimulus increases attention toward that stimulus causing the switch to close less leading to a larger number of pulses being accumulated which results in an overestimation. In contrary, an emotional stimulus presented before the temporal stimuli causes an allocation of resources away from the subsequent temporal stimuli. This causes the switch to flicker closed more leading to a smaller number of pulses being accumulated resulting in an underestimation of time. However, discerning between arousal or attentional affects remains difficult because both can lead to identical behavioral effects [11].

Third, emotion could be causing a bias at the decision-making stage of the model. Previous research by Lieving and colleagues [12] shows that an increased delay between sample stimuli and choice results in a "choose-short" effect. In this situation it is thought that subjects exhibit a bias towards the shorter choice when the retention interval is long because their memory for the stimulus duration decays as the delay increases. In Lieving and colleagues' study they varied the retention interval (RI) by 0, 8, 16, and 32 seconds between and within sessions and found a right-ward shift in the bisection point (BP) for each of the delays compared to the no-delay baseline suggesting that participants experienced a bias to choose short.

Research on emotional timing has utilized emotional faces, words, and sounds as well as other emotional stimuli from the International Affective Picture System (IAPS; for review see: [13]). Previous research found that emotional faces cause an overestimation of time perception. Specifically, it has been shown that angry faces elicit the largest overestimation compared to happy, sad, or neutral expressions with no differences in happy or sad expressions [9, 14–20]. Whereas other studies have shown differential effects of emotion on time. For example, Mioni and colleagues [21, 22] found anger and happiness to cause an overestimation but shame and sadness to cause an underestimation.

In addition to behavioral findings, we examine electrophysiological data using electroencephalography (EEG) recordings. We analyzed event-related potentials (ERPs) associated with timing in the context of emotional stimuli. The contingent negative variation (CNV) is strongly associated with time perception and thought to be an index of timing with the amplitude increasing in relation to the length of the temporal estimates [23, 24]. This ERP was first described by Walter and colleagues [25] as a slow negative wave that occurs between a warning and imperative stimulus. The supplementary motor area (SMA) is thought to be the location where the CNV originates and may represent the pacemaker component of the timing model [26]. Additionally, preceding the CNV a brief negative/positive deflection is often observed (N1/P2) with the N1 component thought to indicate selective attention in that a more negative amplitude indicates increased attention [27, 28] (for review see: [29]). Numerous studies have demonstrated an association between both the N1 and CNV during time estimation [30–32].

The N170/vertex positive potential (VPP) are ERPs associated with face perception and emotion. The VPP is a large positive potential that peaks at the vertex between 140-180ms after the onset of face stimuli [33]. In contrast, the N170 corresponds to the visual N1 component which is the first negative deflection on posterior scalp regions [34]. The N170 has the earliest and strongest difference in amplitude between faces and non-faces with the magnitude being larger for emotional faces, specifically fearful and angry compared to neutral faces [35–37]. The N170 is highly correlated with face selective responses in the temporal lobe (i.e., fusiform gyrus/FFA; [38, 39]).

The late positive component of timing (LPCt) is an additional timing-related ERP associated with decision-making and difficulty in temporal discrimination [40–43]. This ERP presents as a positive frontocentral deflection following the CNV. The time point in which the

LPCt occurs varies depending on the amount of time until response. Some studies show the LPCt at stimulus offset whereas others showing immediately after the subject response (i.e., button press). The aforementioned studies have also found the LPCt to covary with temporal duration and Wiener and Thompson [42] found the LPCt to vary based on choice (short or long).

While previous research has found emotional stimuli to cause an overestimation of time with clear ERPs elicited when perceiving time and viewing emotional stimuli, it is not yet understood which portion of the pacemaker accumulator model might be affected. It could be that emotion is affecting the clock portion of the model via arousal and/or attention or the decision-making portion due to bias. The purpose of this study is an attempt to differentiate between these two possibilities by combining a temporal bisection task with ERP analysis. Whereas previous ERP studies of emotional effects on time perception had subjects measure the duration of emotional stimuli directly, the innovation of the current study is that the stimulus to be timed is not emotional but was instead temporally flanked by the emotional stimuli. The emotional stimuli would have either an emotional valence (negative) or no emotional valence (neutral). The use of this design is twofold: first, when timing an emotional stimulus, the changes in time perception may arise from either clock- or decision-mediated effects but are intermixed during the presentation of the emotional stimulus. Therefore, by flanking the timed stimulus with emotional stimuli before or after the timed stimulus it is possible to determine which stage(s) emotion is affecting. Second, in previous ERP studies of emotion and time perception [43–45], the timing-related ERP components (i.e., N1, CNV) will similarly be intermixed with emotion or face-processing components (i.e., N170, VPP). As such, by temporally separating the emotional stimuli from the timed ones we could account for any independent effects or contributing effects from either response.

We hypothesized that (1) if emotion was affecting the clock portion of the model participants would overestimate the duration when a negative face was shown before the temporal stimulus but (2) if emotion was affecting the decision-making portion of the model participants would overestimate the duration when a negative face was shown after the temporal stimulus. Correspondingly, if emotion was affecting the clock-stage an increase in the CNV would be expected when an emotional stimulus precedes the temporal stimulus (Neg→Neut); whereas if emotion impacted the decision-stage, the response-locked LPCt should be affected when an emotional stimulus follows the temporal stimulus (Neut→Neg). More specifically, if the first hypothesis is supported, we also hypothesize that (1a) if it is arousal that is affecting the clock, we should expect a more negative N170 amplitude when viewing the negative face compared to viewing the neutral face; whereas (1b) if it is attention causing the affect, we should expect a more negative N1 amplitude when viewing the negative face compared to viewing the neutral face.

## Method

### Behavioral study

**Participants.**   Sixty-one participants participated in the study; 14 did not conclude the task and their data were not analyzed. Out of the 47 remaining participants, three were excluded because they always responded "short" independently of stimulus duration. The five other participants were excluded because their performance deviated more than two standard deviations from the BP mean performance. The final sample included 39 young adults (mean age 24.66 years; SD = 3.09; male = 12; 30 right-handed). Participants were recruited through advertisement at the Department of General Psychology at the University of Padova, Italy and students' social media. Participants were not compensated for their participation in this study.

**Materials.** The emotional stimuli included pictures of four different male and four different female faces displaying an angry and a neutral expression selected from the FACES dataset ([46]; Images: 008_y_m_a_b, 008_y_m_n_b, 066_y_m_a_b, 06_y_m_n_b, 010_y_f_a_b, 010_y_f_n_b, 040_y_f_a_b, 040_y_f_n_b). The temporal stimulus was a Gaussian blur presented at the same size and location as the emotional stimuli. PsychoPy2 (v1.90.3; [47]) was used to program the experiment; a link was sent to the participants using the Pavlovia system (www.pavlovia.org). Participants received an email with information regarding the study. Before starting the experimental procedure, they signed the online consent form.

**Procedure.** Participants completed a temporal bisection task on their own computer in which they categorized the duration of a temporal stimulus as either short or long by pressing 'S' or 'L' on the keyboard, respectively. Prior to and following the temporal stimulus, participants saw a human face which expressed either a negative emotion (anger) or a neutral emotion. The face before and after the stimuli was always the same individual and the combinations of faces presented were randomly selected from four male and four female faces from trial to trial. Each trial was classified into one of three conditions: negative first/neutral second (Neg→Neut), neutral first/negative second (Neut→Neg), or neutral first/neutral second (Neut→Neut). The study consisted of two blocks with 168 trials in each. Each trial proceeded as follows: a fixation cross for 1000ms, the first face for 600ms, a 500ms blank screen, the temporal stimulus for one of seven log-spaced durations (300, 360, 433, 520, 624, 749, or 900ms), a 500ms blank screen, the second face for 600ms, a 500ms blank screen, and the word "Respond" until the participant responded by pressing either 'S' or 'L' (Fig 2). Each trial lasted approximately 4–4.6 seconds and each duration was experienced 16 times in each of the three conditions, additionally each block took approximately 15 minutes. Participants did not complete any practice trials and were instructed to judge the duration of the temporal stimulus on each trial as short or long based on all other durations experienced. We used a fixed inter-stimulus interval (ISI) which could lead to subjects using it as a standard in which to compare to the temporal interval; however, if this were the case we would expect the BP to be closer to the ISI (500ms) which is not the case nor would it change the findings of the study. Additionally, we chose the 600ms duration since it corresponded to the bisection point of the intervals used and believed that this would cause the least amount of confusion to participants. We did inform participants that the faces always appeared for the same amount of time so they need not pay attention to the duration of those faces but only focus on the temporal stimulus. If subjects were using the face as a reference, participants would be more likely to say short in the Neg→Neut condition leading to a lower CNV which is not the case. Therefore, we believe that the duration for which the face was presented would not change the results of the study.

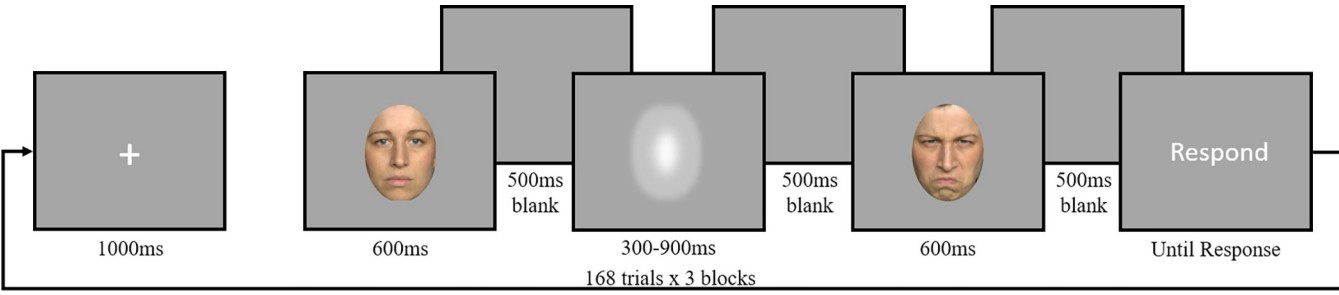

**Fig 2. Study paradigm.** Fixation cross (1000ms), face stimulus (600ms), blank screen (550ms), Gaussian blur (300, 360, 433, 520, 624, 749, or 900ms), blank screen (500ms), face stimulus (600ms), blank screen (500ms), "Respond" (until key press). Example of Neut →Neg trial.

**Data collection/analysis.** The experimental software (PsychoPy2) recorded response time (time from when "Respond" appeared on the screen until the button press) and key presses ('S' or 'L'). We analyzed the bisection point (BP) in each of the possible conditions across the different temporal durations. The BP is derived from the proportion of long responses. Specifically, behavioral data were first analyzed in terms of the proportion of "long" responses of each participant for each of the temporal intervals and for each of the emotional conditions. From this measurement, a psychometric function was fit to each individual subject's data. Each psychometric function consisted of a cumulative Gaussian function fit via a maximum likelihood estimation procedure. From these fits, we then extracted the BP and the standard deviation (σ), which is an estimate of the difference threshold. The BP is an index of perceived duration and can be described as the time value corresponding to the 0.5 probability of "long" responses. We also calculated the Coefficient of Variation (CV) as the standard deviation divided by the BP with higher CVs corresponding to lower temporal sensitivity. Repeated measures ANOVAs were conducted on BP, CV, and RT (reaction time) across conditions (Neg→Neut, Neut→Neg, Neut→Neut) as within-subject factor. RT was also analyzed across temporal intervals.

## EEG study

**Participants.** This study included 31 right-handed participants (18 females, 10 males, 3 undisclosed; 18–29 years old, mean age = 21.7) from the undergraduate student population of George Mason University. All participants had normal or corrected-to-normal vision. Participants were compensated for their time with either research credits or monetary payment. All participants completed a demographic questionnaire and provided written informed consent. All protocols were approved by The Institutional Review Board at George Mason University. Three participants were removed from the analysis because their bisection point was more than two standard deviations from the mean. Another participant was removed due to an issue with the recorded EEG data. Data from a total of 27 participants were analyzed.

**Materials.** The emotional and temporal stimuli were the same as the behavioral only paradigm and the study was completed in person at George Mason University. Participants were seated approximately 36 inches in front of a 32" LCD monitor built by Cambridge Research Systems (120 Hz refresh rate). PsychoPy2 (v1.90.3; [47]) was used to build and run the time bisection task.

**Procedure.** The experimental procedure was the same as the behavioral only study except it consisted of three blocks with 168 trials in each. Again, each trial lasted approximately 4–4.6 seconds and each block took approximately 15 minutes. Participants were given instructions on the task both verbally and written and were instructed to minimize head and eye movements during the trials.

**Data collection/analysis.** Again, we used the recorded response time and key presses to analyze the behavioral data. The EEG data was recorded from 64-actiCAP Slim electrodes connected to an actiCHamp amplifier (Brain Products) using BrainVision Recorder software. The EEG data were sampled at 1000 Hz (n = 22) or 500 Hz (n = 5). As done for the behavioral version of the study, we analyzed the BP, CV, and RT. Repeated measures ANOVAs were conducted on BP, CV, and RT across conditions (Neg→Neut, Neut→Neg, Neut→Neut) as within-subject factor. RT was also analyzed across temporal intervals.

EEG data were sampled at 1000 Hz were down sampled from to 500 Hz and all EEG data were re-referenced to the average of the mastoid channels (T9 and T10) and then epoched from 500ms before to 1000ms after the facial stimuli onset, temporal stimulus onset, and the key press response onset. Independent Component Analysis (ICA) was employed to visually

detect and remove artifacts related to eye blink, muscle, and line noise artifacts [48], after which a 1–50 Hz finite impulse response bandpass filter was applied. Separate epochs were generated for the face stimuli for each of the four face conditions (negative first, neutral first, negative second, neutral second) and for the temporal stimulus for each condition, collapsed across duration (Neg→Neut, Neut→Neg, Neut→Neut), response (short: S, long: L), and condition (Neg→Neut_S, Neg→Neut_L, Neut→Neg_S, Neut→Neg_L, Neut→Neut_S, Neut→Neut_L). All epochs were analyzed using the mean amplitude.

In order to analyze the N170 we calculated the mean amplitude across participants for negative faces and neutral faces at parietal and parieto-occipital electrodes (P7, P8, PO7, PO8) between 145-185ms [49]. For the N1 and CNV, we analyzed the mean amplitude within a frontocentral a-priori cluster of nine electrodes centered on FCz (FCz, Fz, Cz, F1, F2, FC2, C1, C2), within two-time windows: 150-190ms and 250-450ms, respectively [32, 42]. The mean amplitude of the LPCt was analyzed between 200-600ms after the response at the same nine electrodes as the CNV and N1 [42].

## Results

### Behavioral only study

Results showed a significant main effect of condition on the BP ($F_{(2,76)} = 4.13$, $p = 0.020$, $\eta^2 = 0.098$; Table 1; Fig 3). Post-hoc Wilcoxon signed-rank tests showed a lower BP in the Neg→Neut condition compared to Neut→Neut ($W = 575$, $p = 0.009$, $r_{rb} = 0.474$) but not compared to Neut→Neg ($W = 300$, $p = 0.214$, $r_{rb} = -0.231$) indicating a temporal overestimation in the Neg→Neut condition. No differences were observed between Neut→Neg and Neut→Neut ($W = 445$, $p = 0.451$, $r_{rb} = 0.141$). Analysis conducted on CV and RT revealed no significant results ($F_{(2,76)} = 0.08$, $p = 0.926$, $\eta^2 = 0.002$; $F_{(2,76)} = 0.93$, $p = 0.398$, $\eta^2 = 0.024$, respectively).

### EEG study

**Behavioral effects.** Results showed no significant effect of condition on the BP ($F_{(2,52)} = 1.41$, $p = 0.253$, $\eta^2 = 0.052$; Fig 3, Table 1). Although not significant, we observed a minor leftward shift in the BP with a marginally larger shift for Neg→Neut and Neut→Neg, suggesting a slight overestimation of time perception in those conditions (Mean BP: Neg-Neut = 570 ms, Neut-Neg = 574 ms, Neut-Neut = 582 ms). A repeated measures ANOVA was also conducted for CV and for reaction time (RT), two participants were not included in the RT analyses due to missing data caused by an issue with the data files. Neither analysis revealed significant results ($F_{(2,52)} = 0.03$, $p = 0.974$, $\eta^2 = 0.001$; $F_{(2,48)} = 0.02$, $p = 0.811$, $\eta^2 = 0.009$, respectively).

**Electroencephalogram (EEG) effects.** *N170*. A repeated measures ANOVA of the N170 amplitude at posterior electrodes during face presentation revealed a significant main effect of

**Table 1. Behavioral effects.** Displaying mean and standard error for bisection point (BP), contingent variation (CV), and reaction time (RT). Left is behavioral only study results; right is behavioral results from EEG study.

| | | Behavioral Only Study | | | EEG Study | | |
|---|---|---|---|---|---|---|---|
| | | Neut→ Neut | Neg→Neut | Neut→Neg | Neut→ Neut | Neg→Neut | Neut→Neg |
| BP (ms) | *M* | 629.56 | 612.97 | 624.71 | 582.19 | 570.74 | 574.94 |
| | *SE* | 14.83 | 12.81 | 15.18 | 15.07 | 13.99 | 14.50 |
| CV | *M* | 0.23 | 0.23 | 0.24 | 0.30 | 0.29 | 0.25 |
| | *SE* | 0.01 | 0.01 | 0.01 | 0.03 | 0.03 | 0.03 |
| RT (ms) | *M* | 625.44 | 716.01 | 856.00 | 554.85 | 558.19 | 549.63 |
| | *SE* | 57.74 | 68.64 | 210.90 | 49.03 | 49.21 | 49.24 |

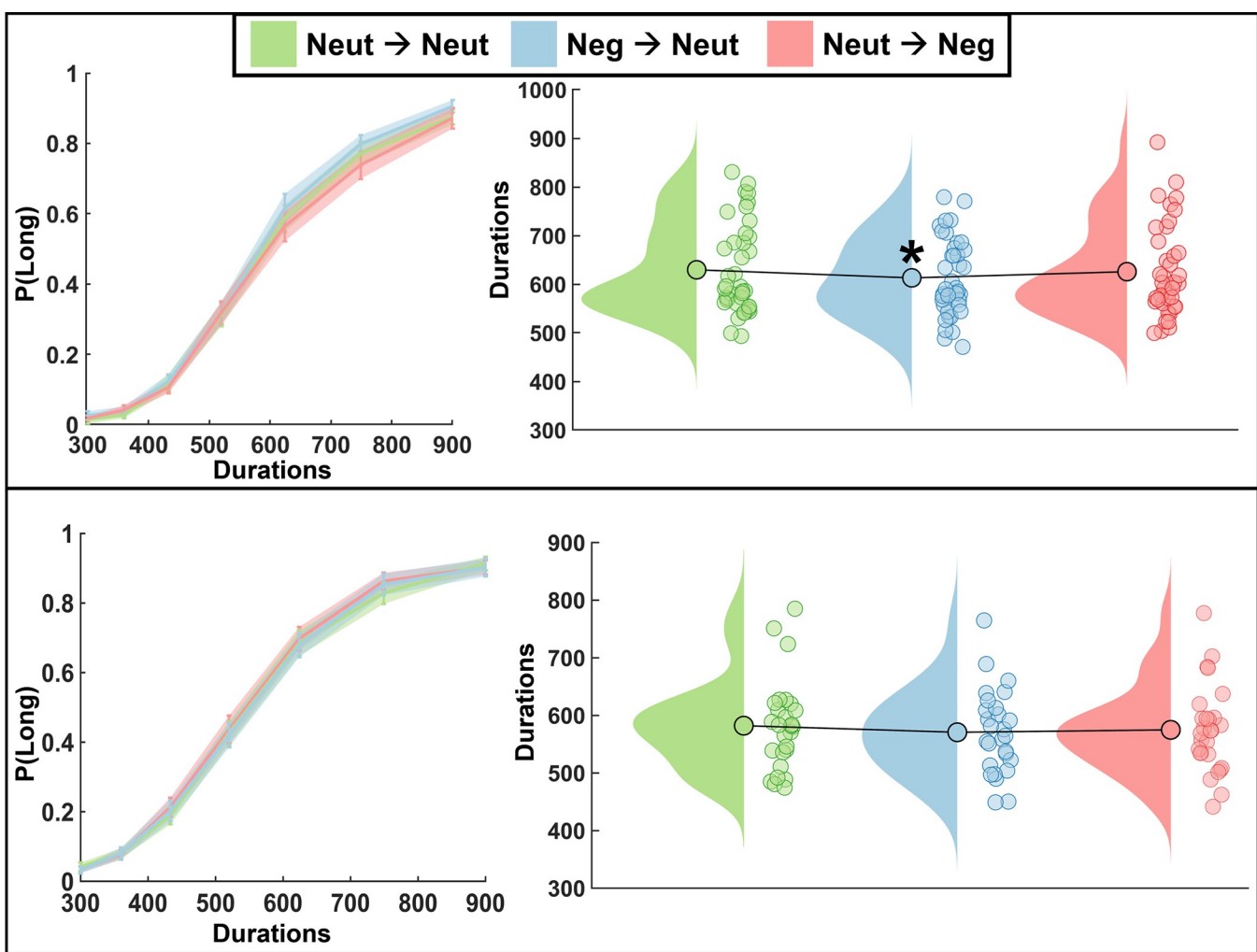

**Fig 3. Behavioral effects.** Psychometric curve (left) displaying the mean proportion of long responses for each duration across condition with standard error bars. Raincloud plot (right) displaying bisection points (BP) for each condition. Top: Behavioral only study results. Significant leftward shift in BP for Neg→Neut condition indicates overestimation of time; Bottom: Behavioral effects from EEG study. Not significant but slight leftward shift in BP suggests minor overestimation of time in Neg→Neut and Neut→Neg. (* = p < 0.05).

emotion, regardless of presentation order ($F_{(1,26)} = 29.76$, $p < .001$, $\eta^2 = 0.199$; $F_{(1,26)} = 1.89$, $p = 0.181$, $\eta^2 = 0.038$, respectively). There was also no interaction effect ($F_{(1,26)} = 0.99$, $p = 0.329$, $\eta^2 = 0.002$). Post-hoc Wilcoxon signed-rank tests showed that the N170 amplitude was significantly larger for negative faces than for neutral faces regardless of whether participants saw the faces before or after the temporal stimulus (Table 2; Fig 4; Face1: $W = 350.00$, $p < .001$, $r_{rb} = 0.852$; Face2: $W = 316.00$, $p < .001$, $r_{rb} = 0.801$). There was also a significant difference when the first face was negative and the second face was neutral ($W = 305.00$, $p = 0.004$, $r_{rb} = 0.614$) but not when the first face was neutral and the second face was negative ($W = 124$, $p = 0.123$, $r_{rb} = -0.344$), this finding could be due to neural suppression.

*N1/CNV.* For the N1 and CNV amplitudes (Table 3; Fig 5), although there was no significant main effect of condition on the N1 ($F_{(2,52)} = 2.09$, $p = 0.13$, $\eta^2 = 0.074$), there was a significant difference in the amplitude between the Neut→Neut condition and the Neg→Neut condition ($W = 101.00$, $p = 0.03$, $r_{rb} = -0.466$) and a marginally significant difference between Neut→Neut and Neut→Neg ($W = 111.00$, $p = 0.06$, $r_{rb} = -0.466$). There was not a significant

**Table 2. N170 results.** Displaying means, standard error, and 95% confidence interval.

| | N170 | | | |
|---|---|---|---|---|
| | Face 1—Negative | Face 1—Neutral | Face 2—Negative | Face 2 Neutral |
| *M* (μV) | 3.31 | 2.11 | 2.72 | 1.74 |
| *SE* | 0.58 | 0.46 | 0.51 | 0.36 |
| 95% CIs | 2.17, 4.45 | 1.12, 2.91 | 1.72, 3.72 | 1.04, 2.44 |

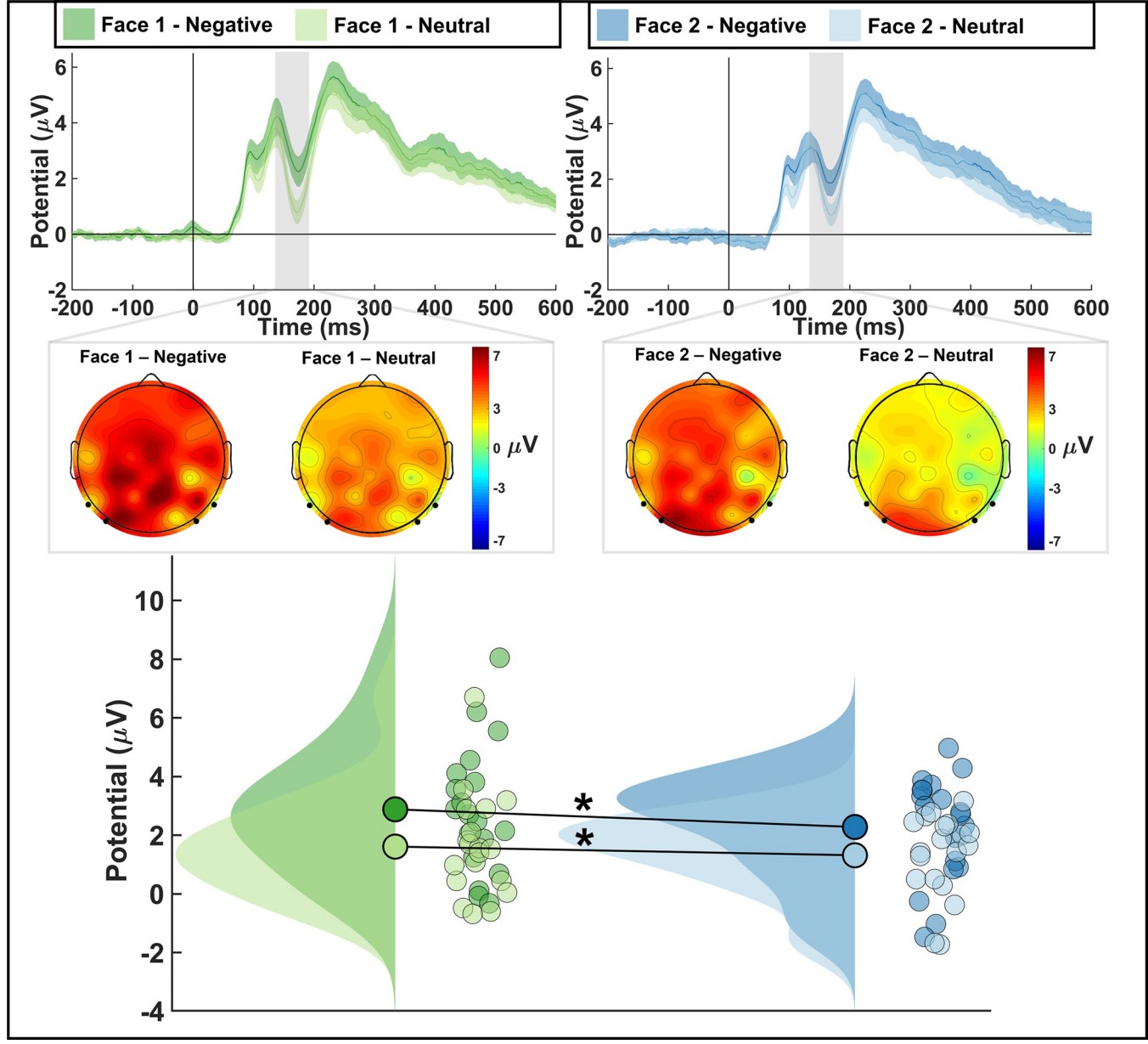

**Fig 4. N170 results.** Left: first face; Right: second face. ERP waveform (top) analyzed at 145-185ms after face stimuli onset at electrodes P7, P8, PO7, PO8, displaying means and standard error. Topography of N170 with electrodes indicated (middle). Raincloud plot (bottom) displaying mean amplitudes. Significant difference between Negative and Neutral faces regardless of order. (* = $p < .001$).

**Table 3. N1 and CNV results.** Displaying means, standard error, and 95% confidence interval (N1-left; CNV-right).

| | N1 | | | CNV | | |
|---|---|---|---|---|---|---|
| | Neut→ Neut | Neg→Neut | Neut→Neg | Neut→ Neut | Neg→Neut | Neut→Neg |
| $M$ (µV) | -1.60 | -2.18 | -2.06 | -2.79 | -3.80 | -3.34 |
| $SE$ | 0.46 | 0.53 | 0.49 | 0.45 | 0.48 | 0.49 |
| 95% CIs | -2.51, -0.69 | -3.21, -1.15 | -3.03, -1.09 | -3.68, -1.91 | -4.74, -2.85 | -4.30, -2.38 |

effect between Neg→Neut and Neut→Neg ($W = 183.00$, $p = 0.90$, $r_{rb} = -0.032$). The CNV, on the other hand, had a significant main effect of condition on the amplitude ($F_{(2,52)} = 5.07$, $p = 0.01$, $\eta^2 = 0.163$) with significant differences between Neg→Neut and Neut→Neut ($W = 71.00$, $p = .004$, $r_{rb} = -0.624$) and Neut→Neg and Neut→Neut ($W = 76.00$, $p = 0.005$, $r_{rb} = -0.598$) but not between Neg→Neut and Neut→Neg ($W = 125$, $p = 0.129$, $r_{rb} = -0.339$).

The significant difference between the Neut→Neg and Neut→Neut conditions in the CNV was an unexpected finding since, at this point in the task, the Neut→Neg and Neut→Neut conditions are identical we did not anticipate there to be a significant difference between them. We therefore opted to average across the Neut→Neut and Neut→Neg conditions for analysis and compare trials in which a negative face came first (Neg1st) with those in which a neutral face came first (Neut1st). A paired samples t-test revealed a significant difference for the CNV ($W = 60.00$, $p = 0.001$, $r_{rb} = -0.683$) but not for the N1 ($W = 154.00$, $p = 0.413$, $r_{rb} = -0.185$). Fig 5 and Table 3 still reflect the initial analysis across the three conditions.

*LPCt.* There was no significant main effect of condition or response on the LPCT amplitude ($F_{(2,52)} = 1.99$, $p = 0.147$, $\eta^2 = 0.038$; $F_{(1,26)} = 1.06$, $p = 0.31$, $\eta^2 = 0.007$, respectively); however, there was an interaction of condition and response (Table 4; Fig 6; $F_{(2,52)} = 3.42$, $p = 0.04$, $\eta^2 = 0.033$). Specifically, there was a significant difference in the LPCt amplitude for short responses between the Neut→Neg and Neut→Neut condition ($W = 324.00$, $p < .001$, $r_{rb} = 0.714$) and a marginally significant difference for short responses between Neg→Neut and Neut→Neg ($W = 266.00$, $p = 0.07$, $r_{rb} = 0.407$). We also found that the LPCt amplitude for short and long responses were significantly different in the Neut→Neut condition only ($W = 303.00$, $p = .005$, $r_{rb} = 0.603$). There were also marginally significant differences between short responses in the Neut→Neut condition and long responses in the Neut→Neg condition as well as between short responses in the Neut→Neg condition and long responses in the Neg→Neut condition.

## Discussion

In the present study, we found that participants overestimated time when an emotional face preceded a timed stimulus, and the EEG results suggest that time was overestimated due to a mixture of arousal (as indexed by the face-responsive N170) and attention (as indexed by the timed stimulus N1). Additionally, an interaction effect of condition and response observed on the LPCt suggests that decision-making was also affected when the emotional face succeeded the temporal stimulus. Together, these findings suggest that emotion can affect both the clock and the decision-making stage of the pacemaker-accumulator model.

The CNV signal has previously been associated with the perception of time [50]. A larger, more negative CNV amplitude has been interpreted to represent a larger number of pulses that have accumulated [32, 51]; therefore, our results suggest that participants overestimated the duration of a neutral stimulus when preceded by an angry face as opposed to a neutral face. However, it remains unclear as to whether the emotional impact on the clock stage is specifically due to attention or arousal as our findings could be interpreted as evidence for both. The face-responsive N170 at posterior electrodes was significantly more negative for angry

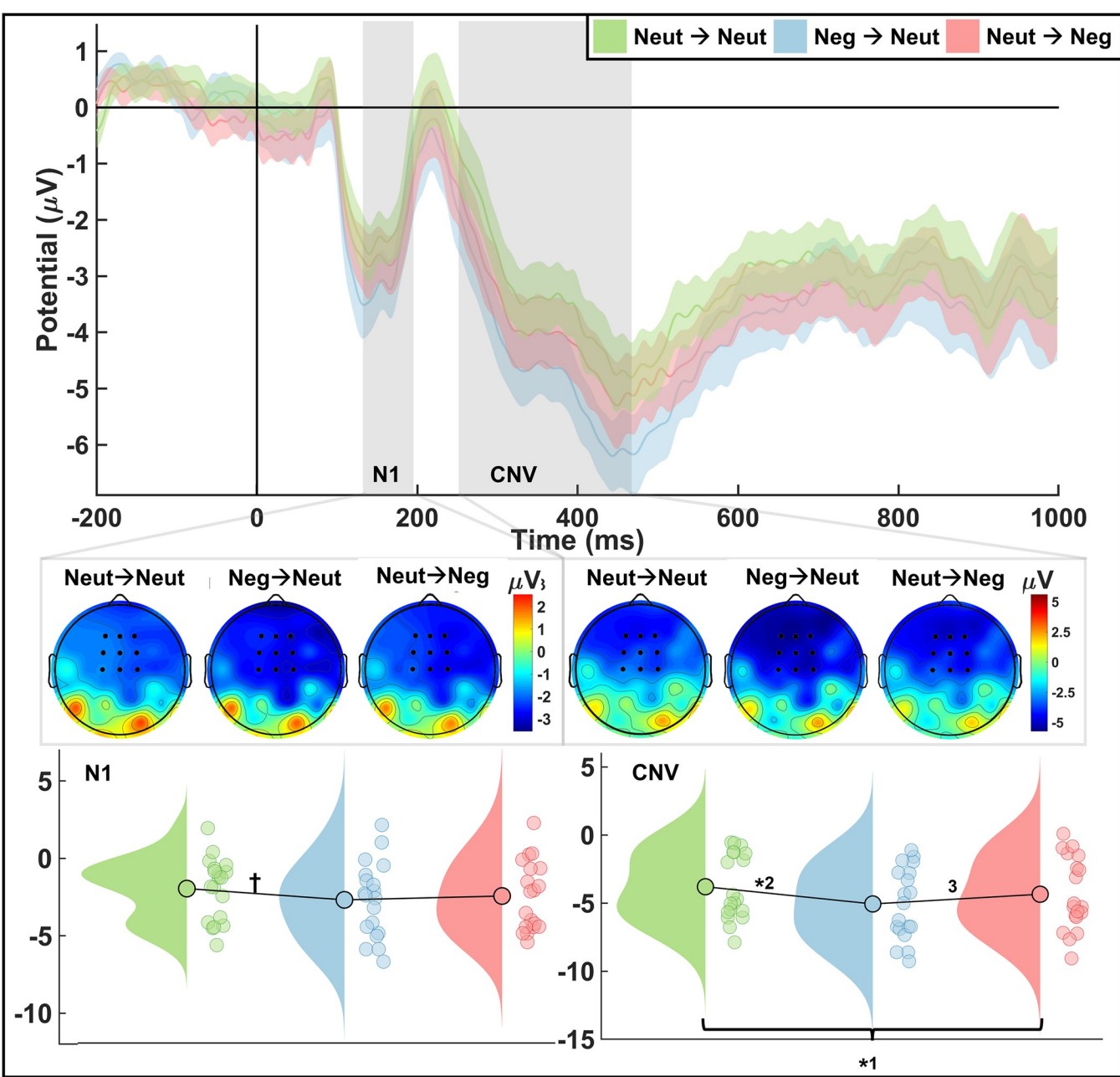

**Fig 5. N1 and CNV results.** ERP waveform (top) displaying means and standard error. Analyzed at 150-190ms and 300-600ms after temporal stimulus onset for N1 and CNV, respectively, at electrodes FCz, Fz, Cz, F1, F2, FC1, FC2, C1, C2. Topography of N1 (left middle) and CNV (right middle) with electrodes indicated. Raincloud plot displaying mean amplitudes for N1 (bottom left) and CNV (bottom right). Marginally significant difference in amplitude for N1 between Neut→Neut and Neg→Neut (p-value: † = 0.06). Significant differences in CNV amplitude between Neg→Neut and Neut→Neut (*2: $p$ = 0.004) as well as Neut→Neg and Neut→Neut (*1: $p$ = 0.005) but not between Neg→Neut and Neut→Neg (3: $p$ = 0.129).

**Table 4. LPCt results.** Displaying means, standard error, and 95% confidence interval for short responses (left) and long responses (right).

| | LPCt (short responses) | | | LPCt (long responses) | | |
|---|---|---|---|---|---|---|
| | Neut→ Neut | Neg→Neut | Neut→Neg | Neut→ Neut | Neg→Neut | Neut→Neg |
| $M$ (μV) | 5.91 | 5.53 | 4.33 | 4.92 | 5.04 | 4.91 |
| SE | 1.56 | 1.60 | 1.18 | 1.49 | 1.22 | 1.27 |
| 95% CIs | -2.51, -0.69 | -3.21, -1.15 | -3.03, -1.09 | -3.68, -1.91 | -4.74, -2.85 | -4.30, -2.38 |

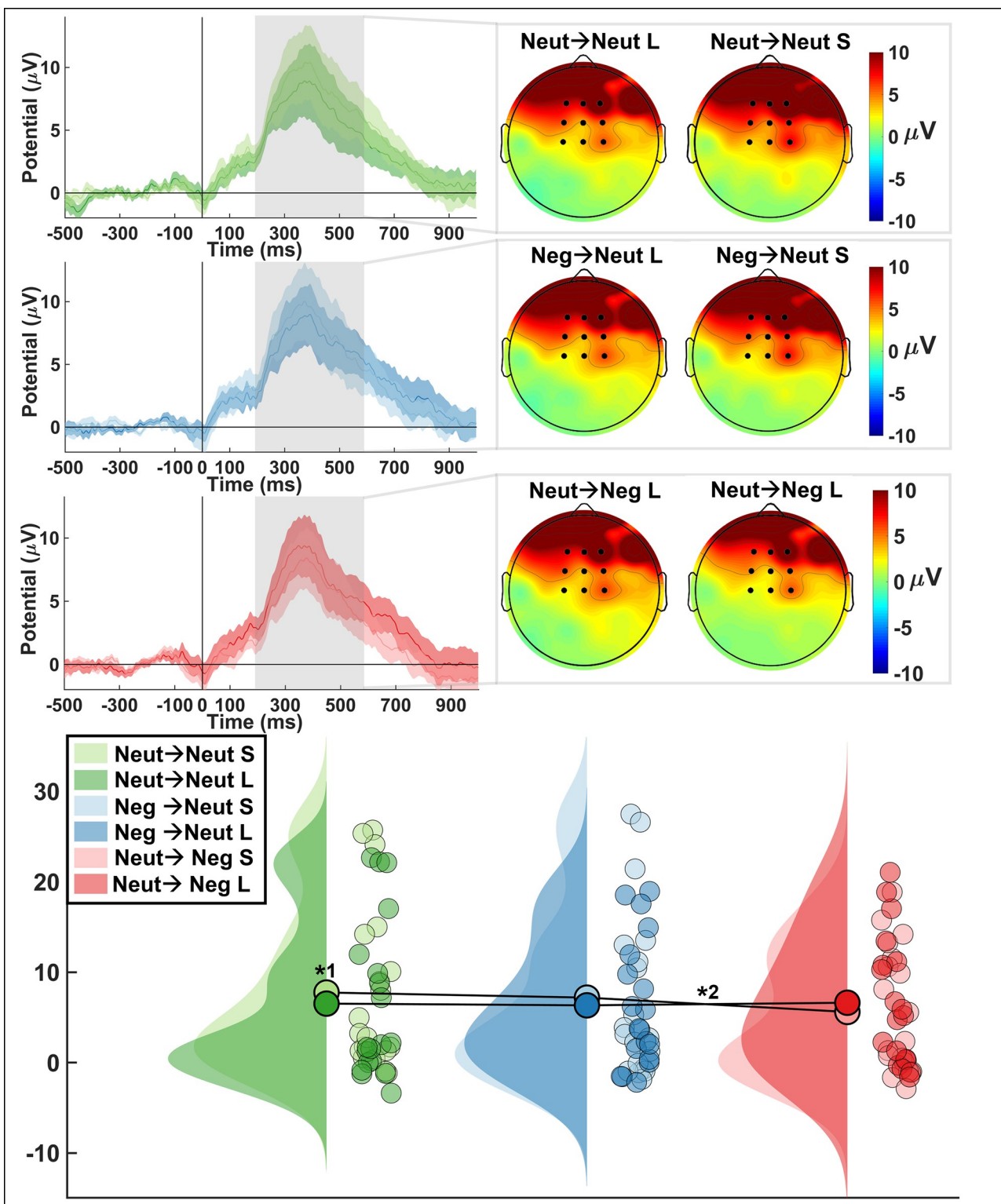

**Fig 6. LPCt results.** ERP waveform (top left) for each of the three conditions and for short and long responses displaying means and standard error. Analyzed at 200-600ms after response onset at electrodes FCz, Fz, Cz, F1, F2, FC1, FC2, C1, C2. Topography of LPCt with electrodes indicated (top right) for each of the three conditions and for short and long responses. Raincloud plot displaying mean amplitudes for the LPCt. Significant difference in amplitude between short and long responses in the Neut→Neut conditions and a significant interaction effect for short responses between the Neut→Neg and Neut→Neut condition. (p-value: *1 = .005; *2 < .001).

faces than for neutral faces suggesting that the angry faces were perceived as more arousing [52]. However, we also found that the N1 response evoked by the subsequent temporal stimulus was marginally larger when preceded by an angry face as opposed to a neutral face suggesting increased attention toward the temporal stimulus [27]. Therefore, overestimation could be due to an increase of the pacemaker rate via arousal, the gate flickering closed less due to increased attention toward the temporal stimulus, or a combination of both. Given previous research and our own findings regarding attentional and arousal effects of emotion on time perception, we suggest that it may not be an either-or process but that both attention and arousal collectively distort time perception in the case of emotional stimuli [11].

The response-locked LPCt revealed that the amplitude varied based on response (short or long) as previously shown by Wiener and Thompson [42]. More notable was an interaction effect between response and condition, with the LPCt amplitude for "short" responses exclusively being significantly lower when participants saw a negative face before responding (Neut→Neg) compared to a neutral face (Neut→Neut). There was also a marginally significant difference in amplitude for "short" responses between Neut→Neut and Neg→Neut suggesting that a negative face before the temporal stimulus still caused some effect at the decision-making stage.

These findings may be related to those that suggest more positive LPCt amplitudes are associated with more difficult discrimination tasks, as evidenced by decreases in accuracy and reaction time associated with higher LPCt amplitudes [53–55]. However, if this was the case, we would expect the LPCt amplitude for both short and long responses to increase by similar amounts in the Neut→Neg condition and only slightly in the Neg→Neut condition compared to Neut→Neut. Instead, the LPCt amplitude for short responses decreased in the Neut→Neg condition only, compared to Neut→Neut, but remains the same for long responses across all conditions. We therefore suggest that the LPCt results instead reveal a bias in decision-making, specifically against choosing "short" as a response option. This finding could be interpreted as a shift in the so-called "choose-short" effect [12], in which humans exhibit a greater tendency to categorize stimuli as "short" in a temporal bisection task. Additionally, Wiener & Thompson [42] found LPCt amplitudes to more closely match choice probability rather than the timed durations as previously thought. Consequently, we believe that the LPCt amplitude difference for short and long responses in the Neut→Neut condition and the change in LPCt amplitude for only short responses in the Neut→Neg condition and less so in the Neg→Neut condition compared to Neut→Neut represents a bias in decision-making. Accordingly, in the control (Neut→Neut) condition, participants are more likely to choose short, evidenced by a larger LPCt for short responses than for long responses, and seeing a negative face prior to responding (Neut→Neg) caused a reduction in this effect. Although the behavioral data does not fully support these conclusions, we believe the EEG results do suggest that the LPCt differences do have something to do with decision-making; however, it is unclear exactly what from these data alone.

We note here that the behavioral data in the EEG study did not reveal significant differences between conditions; however, the behavioral only study did show a significant leftward shift in BP for Neg→Neut compared to Neut→Neg and Neut→Neut suggesting that a negative face before a temporal stimulus resulted in an overestimation of the temporal stimulus. We speculate that the weak behavioral effect in the EEG study may partially be due to the small sample size; however, the emotional stimuli chosen and/or the presentation order of the stimuli may have also contributed. Previous research suggests that emotional faces are not as arousing as other emotional stimuli (e.g. IAPS; [56]), therefore, using facial stimuli may have resulted in a weaker effect [57]. Additionally, the emotional stimuli were presented before or after, rather than during, timing or response. This may have also caused a weaker behavioral effect but was

necessary in order to analyze important ERPs independently from one another. For example, if the emotional stimuli were displayed during the timing or response phase, it would not be possible to fully differentiate the N170 from the N1 and CNV or the LPCt.

Furthermore, the neural components which are more susceptible to these effects than behavior support the behavioral findings. We also note that participants were not explicitly asked to respond as quickly as possible as to whether the duration was short or long. This allowed participants ample time to consider their response, thus decreasing the speed-accuracy tradeoff (i.e., with more time to decide accuracy increases; for review see: [58]. Moreover, previous research has shown that when response time is not constrained participants can change their mind and that changes of mind reliably improve accuracy [59]). We therefore suspect that by not having a time constraint for responding, participants were able to make more accurate decisions and had ample time to change their mind thus improving accuracy. This may explain the lack of a behavioral effect in the Neut→Neg condition even though the LPCt is affected.

In summary, we found that negative emotional faces not only increased the perceived time of an interval but also biased temporal discrimination as evidenced by the CNV and LPCt, respectively, and further supported by the behavioral only results. In addition, our results support that the effect of emotion on time perception may not be exclusively due to arousal or attention alone but rather a combined effect. As far as we are aware, a neural effect of emotional stimuli on decision bias in time estimation has not been observed. As such, we have also provided more evidence for the LPCt as a measure of decision-making bias with more positive amplitudes relating to choice probability.

## Author Contributions

**Conceptualization:** Keri Anne Gladhill, Giovanna Mioni, Martin Wiener.

**Formal analysis:** Keri Anne Gladhill, Giovanna Mioni.

**Investigation:** Keri Anne Gladhill.

**Methodology:** Keri Anne Gladhill, Giovanna Mioni.

**Supervision:** Martin Wiener.

**Visualization:** Keri Anne Gladhill.

**Writing – original draft:** Keri Anne Gladhill.

**Writing – review & editing:** Giovanna Mioni, Martin Wiener.

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
