## [Decision Letter · Decision Letter 0]

3 Jun 2022

PONE-D-22-09658Dissociable Effects of Emotional Stimuli on Electrophysiological Indices of Time and Decision-MakingPLOS ONE

Dear Dr. Gladhill,

Thank you for submitting your manuscript to PLOS ONE. After careful consideration, we feel that it has merit but does not fully meet PLOS ONE’s publication criteria as it currently stands. Therefore, we invite you to submit a revised version of the manuscript that addresses the points raised during the review process especially about the methodological issues.

We look forward to receiving your revised manuscript.

Kind regards,

Francesco Di Russo, Ph.D.

Academic Editor

PLOS ONE

Journal Requirements:

4. We note that Figure 2 includes an image of a participant in the study. 

Reviewers' comments:

Reviewer's Responses to Questions

**Comments to the Author**

1. Is the manuscript technically sound, and do the data support the conclusions?

Reviewer #1: Partly

Reviewer #2: Yes

2. Has the statistical analysis been performed appropriately and rigorously? 

Reviewer #1: Yes

Reviewer #2: Yes

3. Have the authors made all data underlying the findings in their manuscript fully available?

Reviewer #1: Yes

Reviewer #2: Yes

4. Is the manuscript presented in an intelligible fashion and written in standard English?

Reviewer #1: No

Reviewer #2: Yes

5. Review Comments to the Author

Reviewer #1: Thank you for giving me the opportunity to review this paper.

The paper reports a behavioural and an ERP experiment. The behavioural experiment established the influences of emotional faces on the chosen visual temporal task with a relatively larger sample (n=61). The EEG experiment studied the ERPs at different time points with a smaller sample (n=31). The setup is understandable considering the availability of equipment and the difficulty in the experimental setup.

The experiments employed a specific procedure presenting two emotional faces in a trial, before and after the visual temporal stimuli. The emotional faces before the stimuli were expected to influence attention and arousal to the perceptual period, while the faces after the stimuli provided information on how they affect the decision process. It is a novel setup to me.

The authors shall address the following concerns before the manuscript shall be considered for publication.

1. The authors shall provide a more detailed account of the data exclusion criteria. On page 11, “three participants were removed because their behavioural data was more than two standard deviations from the mean”, but it does not mention which parameters. A similar comment applied to the participant removed due to ERP characteristics that it is unclear which components or just the grand average. Further, the two participants excluded in the RT analyses (page 14) also required further elaboration.

2. On page 12, the authors reported using ICA to detect and remove artefacts relating to eye blinks and noise. A more detailed description to the cleaning pipeline will be appreciated, and please cite the references if the authors followed any published protocols.

3. Please provide detailed descriptions of the instructions to the participants. For example, it is unclear how the participants understand the ‘long’ and ‘short’ responses. Is there any practice block?

4. Please provide the measurement units to the tables. Should it be milliseconds (ms) for BP and RT?

5. Please provide effect sizes to the behavioural effects of the EEG study though they are not significant.

6. Kindly elaborate on how the amplitude analyses were done. For example, do the authors use local peak or mean amplitude in the EPOCH?

7. It would be nice to have a EEG map to show the chosen electrodes for the ERP analyses of various components. Further, there appears to be an overlap of different ERPs in time. It would be nice if there is a timeline in a similar format to Figure 2 but shows the duration of different EPOCHs. It is also unclear how the potential overlap may influence the ERP analyses. The authors shall critically discuss this possibility.

8. The authors describe arousal and attention influence the pacemaker and the gate, respectively, in affecting the visual temporal task. A more critical analysis will be appreciated. Further, on page 19, the authors describe the “gate opening wider via increased attention”. Instead of increased attention, it appears to be more common to describe attention as being diverted. Also, the gate is described as flickered or delayed in its opening instead of opening wider.

9. The argument in the discussion about LPCt appears to be weak. It contrasts the LPCt amplitudes between the Neut>Neg and Neut>Neut condition and claims a marginally significant difference in the amplitudes among the ‘short’ responses. However, there is no behavioural difference between these conditions in Experiments 1 and 2. There is, perhaps, insufficient support for the effect of negative faces on the decision process stage.

10. Partially follow up on #3, the authors shall probably acknowledge the limitation of using a fixed ISI (500ms) similar to their physical bisection point. Depending on the experimental protocols, the participants may use the ISI to help the judgement.

Reviewer #2: The aim of the present study is to determine whether the effect of emotion on time perception results from an alteration of the clock speed or from a decision bias. To distinguish between the two hypotheses, neutral or negative (angry) faces were presented before and/or after the visual stimulus whose duration was evaluated by participants. Two experiments were conducted. A first behavioural experiment revealed that the stimulus duration was overestimated when preceded by an angry face in line with an increase of the clock speed. A second EEG experiment revealed that the contingent negative variation (CNV) measured during the visual stimulus was enhanced when preceded by an angry face still in line with an increase of the clock speed. Furthermore, the late positive component of timing (LPCt) following 'short' responses was reduced when an angry face was presented after the visual stimulus suggesting that decision processes were also altered by emotion.

This is a well-designed study based on the combination of two approaches (behavioural and EEG) addressing a straightforward issue. Although results are not as clear as hypotheses, I would suggest that the paper would be publishable as it proposes an original paradigm allowing to carefully examine the effect of emotion on time processing which would aid in a better understanding of the underlying neural and cognitive mechanisms. However, I have several concerns with the results and the way they are discussed as well as with some aspects of the methodology. Here are my comments and suggestions:

Major Points:

1. As noted by the authors, in the EEG experiment, ERPs measured during the visual stimulus were expected to differ between the neg-neut and the neut-neut conditions but not between the neut-neg and the neut-neut conditions, since at this moment of the trial, the two conditions are identical. However, the CNV amplitude was significantly larger and the N1 amplitude tended to be larger for neut-neg trials than for neut-neut trials. The authors proposed that the larger CNV amplitude for the neut-neg condition would only concern trials following a neg-neut trial because after a neg-neut trial, participants would predict a negative face in first position and thus, a neutral face presented in first position would produce a positive prediction error which would induce an increased CNV amplitude. However, the same logic seems to apply for both neut-neg and neut-neut trials following a neg-neut trial (note also that, according to this explanation in terms of predictive error, a negative face presented in first position should produce a negative prediction error and thus and a duration underestimation and a lower CNV amplitude). Alternatively, the authors proposed that, after a neg-neut trial, when a neutral face is presented in first position, participants would anticipate a negative face in second position and that such anticipation would induce an increased CNV amplitude. However, here again, the same reasoning seems to apply equally for both neut-neg and neut-neut trials. These different explanations thus appear as confusing. As there is no expected difference between neut-neg and neut-neut conditions for the N1 and the CNV measured during the visual stimulus, could these two conditions be grouped together and the statistical analyses be performed between trials starting with a negative face vs trials starting with a neutral face?

2. The paradigm is not symmetric due to the absence of neg-neg condition. Thus, when the first face is negative, participants can predict than the second face will be neutral as there is not neg-neg trial. By contrast, when the first face is neutral, participants cannot anticipate whether the second face will be neutral of negative. Participants would thus be in a "predictive context" following a negative face and a in an "unpredictive context" following a neutral face. How exclude the fact that the longer duration estimates and the larger CNV amplitude observed for a visual stimulus preceded by an angry face would be due to the context (predictive vs unpredictive) rather than to the emotion per se (angry vs neutral)?

Minor Points:

1. Concerning the first behavioural experiment, how explain the important loss of participants from the inclusion (61) to the statistical analyses (39 participants)? It is not clear whether this behavioural experiment was an online experiment and whether participants performed the task at home in an uncontrolled context. A bisection task was used but there is no information about the anchors (300 and 900ms?) and the number of presentations of each of them.

2. Faces were presented during 600ms which corresponds approximately to the bisection point. Why did you use this duration? Is it possible that participants actually compared the duration of the visual stimulus with that of the faces rather than with that of the memorized anchors?

3. Although no significant effect of condition on the bisection point (BP) was observed in the EEG experiment, did you examine and observe correlations between behavioural and ERP indices? For example, is the difference of BP between neg-neut and neut-neut conditions correlated with the difference of CNV amplitude between the two same conditions? Did you also examine and observe correlations between the different ERP indices? For example, is the difference of N170 amplitude between negative and neutral faces presented in first position correlated with the difference of CNV amplitudes between trials starting with a negative face and trials starting with a neutral face?

4. There is a problem with the sentence in lines 307 to 310. Concerning the results on the N170, how explain that amplitudes are larger for faces presented in second position than for faces presented in first position? Could this be due to neural suppression by repetition of the same face?

5. The sentence in lines 107 to 109 (marking the beginning of a new section concerning the investigation of time perception with EEG) could be placed at the beginning of the following paragraph.

6. PLOS authors have the option to publish the peer review history of their article (what does this mean?). If published, this will include your full peer review and any attached files.

Reviewer #1: **Yes: **Li Wang On

Reviewer #2: No

---

## [Author Response · Author response to Decision Letter 0]

2 Aug 2022

We thank the reviews for their helpful comments in the improvement of our manuscript. We have addressed each comment individually and believe that it has enhanced the organization and understanding of our paper. Please see below for individual responses in blue and see the manuscript for changes and corrections which are indicated by red text.

Journal Requirements:

We have addressed the formatting requirements as displayed in the PLOS ONE style templates.

We will provide the necessary information for the data repository upon acceptance and understand that it will be held for publication until it has been provided.

We have verified that the methods section includes the full ethics statement as well as the full name of the IRB committee that approved our study.

4. We note that Figure 2 includes an image of a participant in the study. 

We appreciate the concern regarding figure 2; however, we want to clarify that the image being referred to in figure 2 is an example of the face stimuli that was used in the paradigm. The source of these images have been cited in the manuscript.

Reviewers' comments:

Reviewer's Responses to Questions

Comments to the Author

1. Is the manuscript technically sound, and do the data support the conclusions?

Reviewer #1: Partly

Reviewer #2: Yes

2. Has the statistical analysis been performed appropriately and rigorously?

Reviewer #1: Yes

Reviewer #2: Yes

3. Have the authors made all data underlying the findings in their manuscript fully available?

Reviewer #1: Yes

Reviewer #2: Yes

4. Is the manuscript presented in an intelligible fashion and written in standard English?

Reviewer #1: No

Reviewer #2: Yes

5. Review Comments to the Author

Please use the space provided to explain your answers to the questions above. You may also include additional comments for the author, including concerns about dual publication, research ethics, or publication ethics. 

Reviewer #1: Thank you for giving me the opportunity to review this paper.

The paper reports a behavioural and an ERP experiment. The behavioural experiment established the influences of emotional faces on the chosen visual temporal task with a relatively larger sample (n=61). The EEG experiment studied the ERPs at different time points with a smaller sample (n=31). The setup is understandable considering the availability of equipment and the difficulty in the experimental setup.

The experiments employed a specific procedure presenting two emotional faces in a trial, before and after the visual temporal stimuli. The emotional faces before the stimuli were expected to influence attention and arousal to the perceptual period, while the faces after the stimuli provided information on how they affect the decision process. It is a novel setup to me.

The authors shall address the following concerns before the manuscript shall be considered for publication.

1. The authors shall provide a more detailed account of the data exclusion criteria. On page 11, “three participants were removed because their behavioural data was more than two standard deviations from the mean”, but it does not mention which parameters. A similar comment applied to the participant removed due to ERP characteristics that it is unclear which components or just the grand average. Further, the two participants excluded in the RT analyses (page 14) also required further elaboration.

We thank the reviewer for bringing our attention to the lack of detail in regard to participant exclusions. We have addressed the issues in the manuscript. For the behavioral data we added additional information regarding which parameter led to the exclusion. We additionally added more detail about the ERP data exclusions. As for the RT analysis we included the following information “...due to missing data caused by an issue with the data files.” to further elaborate the reason for exclusion from the analysis.

2. On page 12, the authors reported using ICA to detect and remove artefacts relating to eye blinks and noise. A more detailed description to the cleaning pipeline will be appreciated, and please cite the references if the authors followed any published protocols.

Thank you for this suggestion, we have included in the text that artifacts were visually inspected and removed as well as included the citation Jung et al., 1997..

3. Please provide detailed descriptions of the instructions to the participants. For example, it is unclear how the participants understand the ‘long’ and ‘short’ responses. Is there any practice block?

We have addressed this issue by adding in the additional detail regarding instructions participants received and whether or not there was a practice block. We added the line “ Participants did not complete any practice trials and were instructed to judge the duration of the temporal stimulus on each trial as short or long based on all other durations experienced.” in the methods section of the behavioral study.

4. Please provide the measurement units to the tables. Should it be milliseconds (ms) for BP and RT?

We appreciate the reviewer pointing out this oversight. We have addressed the issue and the tables now reflect units of measurement.

5. Please provide effect sizes to the behavioural effects of the EEG study though they are not significant.

This has been addressed and effect sizes have been included in the manuscript.

6. Kindly elaborate on how the amplitude analyses were done. For example, do the authors use local peak or mean amplitude in the EPOCH?

We added the statement “All epochs were analyzed using the mean amplitude.” to the methods section of the manuscript.

7. It would be nice to have a EEG map to show the chosen electrodes for the ERP analyses of various components. Further, there appears to be an overlap of different ERPs in time. It would be nice if there is a timeline in a similar format to Figure 2 but shows the duration of different EPOCHs. It is also unclear how the potential overlap may influence the ERP analyses. The authors shall critically discuss this possibility.

Thank you for your input, we have added an electrode map of chosen electrodes to the topography maps in figures 4, 5, and 6. As for the epoch overlaps, the only overlap would be for the CNV ERP for the two shortest durations; however, the time frame of each epoch we analyzed does not overlap with any other epoch time frame. We therefore believe that there is no issue to address in regards to the possibility of overlap in the ERP analysis.

8. The authors describe arousal and attention influence the pacemaker and the gate, respectively, in affecting the visual temporal task. A more critical analysis will be appreciated. Further, on page 19, the authors describe the “gate opening wider via increased attention”. Instead of increased attention, it appears to be more common to describe attention as being diverted. Also, the gate is described as flickered or delayed in its opening instead of opening wider.

We thank the reviewer for bringing our attention to this issue. We have added in more text in regard to the pacemaker or gate influences. We also corrected the issue in the discussion about the gate opening wider to be more clear.

9. The argument in the discussion about LPCt appears to be weak. It contrasts the LPCt amplitudes between the Neut>Neg and Neut>Neut condition and claims a marginally significant difference in the amplitudes among the ‘short’ responses. However, there is no behavioural difference between these conditions in Experiments 1 and 2. There is, perhaps, insufficient support for the effect of negative faces on the decision process stage.

We agree with the review that the LPCt discussion is weak; however we do believe it is worth mentioning the effect we observed and potential reasons for that difference. In order to address this issue we added the following text to the end of the LPCt discussion. “Although the behavioral data does not fully support these conclusions, we believe the EEG results do suggest that the LPCt differences do have something to do with decision-making; however, it is unclear exactly what from these data alone.”

10. Partially follow up on #3, the authors shall probably acknowledge the limitation of using a fixed ISI (500ms) similar to their physical bisection point. Depending on the experimental protocols, the participants may use the ISI to help the judgement.

We have addressed this issue in the methods section of the text with the following statement. “We used a fixed inter-stimulus interval (ISI) which could lead to subjects using it as a standard in which to compare the temporal interval to; however, if this were the case we would expect the BP to be closer to the ISI (500ms) which is not the case nor would it change the findings of the study.”

Reviewer #2: The aim of the present study is to determine whether the effect of emotion on time perception results from an alteration of the clock speed or from a decision bias. To distinguish between the two hypotheses, neutral or negative (angry) faces were presented before and/or after the visual stimulus whose duration was evaluated by participants. Two experiments were conducted. A first behavioural experiment revealed that the stimulus duration was overestimated when preceded by an angry face in line with an increase of the clock speed. A second EEG experiment revealed that the contingent negative variation (CNV) measured during the visual stimulus was enhanced when preceded by an angry face still in line with an increase of the clock speed. Furthermore, the late positive component of timing (LPCt) following 'short' responses was reduced when an angry face was presented after the visual stimulus suggesting that decision processes were also altered by emotion.

This is a well-designed study based on the combination of two approaches (behavioural and EEG) addressing a straightforward issue. Although results are not as clear as hypotheses, I would suggest that the paper would be publishable as it proposes an original paradigm allowing to carefully examine the effect of emotion on time processing which would aid in a better understanding of the underlying neural and cognitive mechanisms. However, I have several concerns with the results and the way they are discussed as well as with some aspects of the methodology. Here are my comments and suggestions:

Major Points:

1. As noted by the authors, in the EEG experiment, ERPs measured during the visual stimulus were expected to differ between the neg-neut and the neut-neut conditions but not between the neut-neg and the neut-neut conditions, since at this moment of the trial, the two conditions are identical. However, the CNV amplitude was significantly larger and the N1 amplitude tended to be larger for neut-neg trials than for neut-neut trials. The authors proposed that the larger CNV amplitude for the neut-neg condition would only concern trials following a neg-neut trial because after a neg-neut trial, participants would predict a negative face in first position and thus, a neutral face presented in first position would produce a positive prediction error which would induce an increased CNV amplitude. However, the same logic seems to apply for both neut-neg and neut-neut trials following a neg-neut trial (note also that, according to this explanation in terms of predictive error, a negative face presented in first position should produce a negative prediction error and thus and a duration underestimation and a lower CNV amplitude). Alternatively, the authors proposed that, after a neg-neut trial, when a neutral face is presented in first position, participants would anticipate a negative face in second position and that such anticipation would induce an increased CNV amplitude. However, here again, the same reasoning seems to apply equally for both neut-neg and neut-neut trials. These different explanations thus appear as confusing. As there is no expected difference between neut-neg and neut-neut conditions for the N1 and the CNV measured during the visual stimulus, could these two conditions be grouped together and the statistical analyses be performed between trials starting with a negative face vs trials starting with a neutral face?

We agree with the reviewer and have corrected the manuscript to reflect the suggestion. We opted to instead compare between trials starting with a negative face first with those in which a neutral face came first. Results of the analysis have been updated in the manuscript; however, we are leaving the tables and figures as initially analyzed for completeness. 

2. The paradigm is not symmetric due to the absence of neg-neg condition. Thus, when the first face is negative, participants can predict than the second face will be neutral as there is not neg-neg trial. By contrast, when the first face is neutral, participants cannot anticipate whether the second face will be neutral of negative. Participants would thus be in a "predictive context" following a negative face and a in an "unpredictive context" following a neutral face. How exclude the fact that the longer duration estimates and the larger CNV amplitude observed for a visual stimulus preceded by an angry face would be due to the context (predictive vs unpredictive) rather than to the emotion per se (angry vs neutral)?

We agree that the addition of a neg-neg condition could have been beneficial to the study; however, we elected not to include it as we were only looking at the effects of a singular negative face before and after a stimulus rather than all possible combinations. We would like to note that if participants were in a predictive context following a negative face we would expect the CNV to be lower rather than higher since there is no anticipation of an upcoming negative face, which is not the case. Therefore, we believe the paradigm to be sufficient for the research question in this case. 

Minor Points:

1. Concerning the first behavioural experiment, how explain the important loss of participants from the inclusion (61) to the statistical analyses (39 participants)? It is not clear whether this behavioural experiment was an online experiment and whether participants performed the task at home in an uncontrolled context. A bisection task was used but there is no information about the anchors (300 and 900ms?) and the number of presentations of each of them.

We thank the reviewer for bringing our attention to this issue. Within the text we explain that 14 participants did not complete the study, 3 were excluded due to always responding short despite no matter the duration, and 5 were excluded due to BP’s being more than two standard deviations from the mean. We have clarified in the text that this was an online study without compensation and we believe this led to the large number not completing the study as well as the poor performance. As for the anchors, we provided no explicit anchors and instead used the “partition” method employed by our lab and others (cf. Wearden & Ferrera, 1995) in which subjects simply judge each interval relative to all previously experienced intervals; prior work has demonstrated identical performance on this version of the task to those with explicit anchors.

Wearden, J. H., & Ferrara, A. (1995). Stimulus spacing effects in temporal bisection by humans. The Quarterly Journal of Experimental Psychology, 48(4), 289-310.

2. Faces were presented during 600ms which corresponds approximately to the bisection point. Why did you use this duration? Is it possible that participants actually compared the duration of the visual stimulus with that of the faces rather than with that of the memorized anchors?

We thank the reviewer for questioning the duration of faces; however, we chose the 600ms duration because it corresponded to the bisection point. We believe that, since a duration had to be chosen, this would cause the least amount of confusion to the participants. In the verbal instructions participants were informed that the faces always appeared for the same amount of time (but not the temporal stimulus) so they need not pay attention to their duration but rather only focus on the temporal stimulus and whether or not it was short or long based on all other durations experienced. However, if subjects were using the face as a reference we would expect that in the neg-neut condition participants would be more likely to say short which would lead to a lower CNV because the face itself would appear longer making the temporal interval appear relatively shorter. We therefore believe that the interval for which the face presented would change the results of the study.. 

3. Although no significant effect of condition on the bisection point (BP) was observed in the EEG experiment, did you examine and observe correlations between behavioural and ERP indices? For example, is the difference of BP between neg-neut and neut-neut conditions correlated with the difference of CNV amplitude between the two same conditions? Did you also examine and observe correlations between the different ERP indices? For example, is the difference of N170 amplitude between negative and neutral faces presented in first position correlated with the difference of CNV amplitudes between trials starting with a negative face and trials starting with a neutral face?

We appreciate the reviewer’s suggestion to consider additional analyses such as correlations between the behavioral data and ERP indices. We did look into such correlations but unfortunately did not observe any significant correlations between behavioral data and ERP indices nor were there correlations between the ERP indices.

4. There is a problem with the sentence in lines 307 to 310. Concerning the results on the N170, how explain that amplitudes are larger for faces presented in second position than for faces presented in first position? Could this be due to neural suppression by repetition of the same face?

We have made the necessary corrections to the referred sentences and suggest that this finding could be due to neural suppression. 

5. The sentence in lines 107 to 109 (marking the beginning of a new section concerning the investigation of time perception with EEG) could be placed at the beginning of the following paragraph.

We agree with the reviewer that this sentence would serve better as the beginning of the next paragraph and have made this change in the manuscript.

6. PLOS authors have the option to publish the peer review history of their article (what does this mean?). If published, this will include your full peer review and any attached files.

Do you want your identity to be public for this peer review? For information about this choice, including consent withdrawal, please see our Privacy Policy.

Reviewer #1: Yes: Li Wang On

Reviewer #2: No

---

## [Decision Letter · Decision Letter 1]

19 Sep 2022

PONE-D-22-09658R1Dissociable effects of emotional stimuli on electrophysiological indices of time and decision-makingPLOS ONE

Dear Dr. Gladhill,

Thank you for submitting your manuscript to PLOS ONE. After careful consideration, we feel that it has merit but does not fully meet PLOS ONE’s publication criteria as it currently stands. Therefore, we invite you to submit a revised version of the manuscript that addresses the points raised during the review process.

We look forward to receiving your revised manuscript.

Kind regards,

Michael B. Steinborn, PhD

Section Editor

PLOS ONE

Journal Requirements:

Reviewers' comments:

Reviewer's Responses to Questions

**Comments to the Author**

1. If the authors have adequately addressed your comments raised in a previous round of review and you feel that this manuscript is now acceptable for publication, you may indicate that here to bypass the “Comments to the Author” section, enter your conflict of interest statement in the “Confidential to Editor” section, and submit your "Accept" recommendation.

Reviewer #1: All comments have been addressed

2. Is the manuscript technically sound, and do the data support the conclusions?

Reviewer #1: Yes

3. Has the statistical analysis been performed appropriately and rigorously? 

Reviewer #1: Yes

4. Have the authors made all data underlying the findings in their manuscript fully available?

Reviewer #1: Yes

5. Is the manuscript presented in an intelligible fashion and written in standard English?

Reviewer #1: Yes

6. Review Comments to the Author

Reviewer #1: The authors have addressed most comments. Both reviewers asked questions about presenting fixed durations before the target (i.e. a 600ms face and a 500ms blank screen). Another common concern was the lack of practice and explicit anchors. The authors’ responses are satisfactory. They have included some elaborations in the revised manuscript while leaving out some in the reply letter only. A brief paragraph discussing these methodological issues will be appreciated.

There are two minor issues in the figures.

1. The legends in the new topography maps overlap and cannot be read clearly (Figure 4 and 5).

2. Figure 5 caption describes *1 and *2, but the figure shows *2 and *3.

7. PLOS authors have the option to publish the peer review history of their article (what does this mean?). If published, this will include your full peer review and any attached files.

Reviewer #1: **Yes: **Wang On LI

---

## [Author Response · Author response to Decision Letter 1]

20 Sep 2022

PONE-D-22-09658R1

Dissociable effects of emotional stimuli on electrophysiological indices of time and decision-making

PLOS ONE

We thank the reviewers for their comments. We have addressed each comment individually in blue text. Please see the manuscript for changes indicated by red text.

Journal Requirements:

We have reviewed are reference list and believe it to be complete and correct.

Reviewers' comments:

Reviewer's Responses to Questions

Comments to the Author

1. If the authors have adequately addressed your comments raised in a previous round of review and you feel that this manuscript is now acceptable for publication, you may indicate that here to bypass the “Comments to the Author” section, enter your conflict of interest statement in the “Confidential to Editor” section, and submit your "Accept" recommendation.

Reviewer #1: All comments have been addressed

2. Is the manuscript technically sound, and do the data support the conclusions?

Reviewer #1: Yes

3. Has the statistical analysis been performed appropriately and rigorously?

Reviewer #1: Yes

4. Have the authors made all data underlying the findings in their manuscript fully available?

Reviewer #1: Yes

5. Is the manuscript presented in an intelligible fashion and written in standard English?

Reviewer #1: Yes

6. Review Comments to the Author

Reviewer #1: The authors have addressed most comments. Both reviewers asked questions about presenting fixed durations before the target (i.e. a 600ms face and a 500ms blank screen). Another common concern was the lack of practice and explicit anchors. The authors’ responses are satisfactory. They have included some elaborations in the revised manuscript while leaving out some in the reply letter only. A brief paragraph discussing these methodological issues will be appreciated.

We thank the reviewer for drawing our attention to the fact that we did not include these discussions in the manuscript itself. We have corrected this issue by adding the following to the methods section: “Additionally, we chose the 600ms duration since it corresponded to the bisection point of the intervals used and believed that this would cause the least amount of confusion to participants. We did inform participants that the faces always appeared for the same amount of time so they need not pay attention to the duration of those faces but only focus on the temporal stimulus. If subjects were using the face as a reference, participants would be more likely to say short in the NegàNeut condition leading to a lower CNV which is not the case. Therefore, we believe that the duration for which the face was presented would not change the results of the study.”

There are two minor issues in the figures.

1. The legends in the new topography maps overlap and cannot be read clearly (Figure 4 and 5).

We thank the reviewer for catching this formatting issue. We have fixed the figures and uploaded corrected ones.

2. Figure 5 caption describes *1 and *2, but the figure shows *2 and *3.

We appreciate the reviewers attention to detail and catching this error. We have updated the figure caption with the correct information.

7. PLOS authors have the option to publish the peer review history of their article (what does this mean?). If published, this will include your full peer review and any attached files.

Do you want your identity to be public for this peer review? For information about this choice, including consent withdrawal, please see our Privacy Policy.

Reviewer #1: Yes: Wang On LI

---

## [Editor Report · Decision Letter 2]

2 Oct 2022

Dissociable effects of emotional stimuli on electrophysiological indices of time and decision-making

PONE-D-22-09658R2

Dear Dr. Gladhill,

We’re pleased to inform you that your manuscript has been judged scientifically suitable for publication and will be formally accepted for publication once it meets all outstanding technical requirements.

Kind regards,

Michael B. Steinborn, PhD

Section Editor

PLOS ONE
---

## [Editor Report · Acceptance letter]

9 Nov 2022

PONE-D-22-09658R2 

Dissociable effects of emotional stimuli on electrophysiological indices of time and decision-making 

Dear Dr. Gladhill:

I'm pleased to inform you that your manuscript has been deemed suitable for publication in PLOS ONE. Congratulations! Your manuscript is now with our production department. 

Kind regards, 

on behalf of

Dr. Michael B. Steinborn 

Section Editor

PLOS ONE